# Effect of Pulse Frequency on the Microstructure and the Degradation of Pulse Electroformed Zinc for Fabricating the Shell of Biodegradable Dosing Pump

**DOI:** 10.3390/bioengineering9070289

**Published:** 2022-06-29

**Authors:** Shuhui Wu, Yizhuo Luo, Wei Hu, Yonghong Chen, Zhi Huang

**Affiliations:** 1Institute of Biomedical Engineering, School of Basic Medical Sciences, Central South University, Changsha 410017, China; wushuhui@csu.edu.cn (S.W.); 8208191623@csu.edu.cn (Y.L.); 8208191624@csu.edu.cn (W.H.); 2Orthopedic and Traumatology Department, Chenzhou 3rd People’s Hospital, Chenzhou 423000, China; gwstah@gmail.com

**Keywords:** zinc, coating, bioabsorbable, electroform

## Abstract

In this work, we applied single-pulse electrodeposition method to prepare biodegradable zinc coating for the shell of an implantable dosing pump, and explored the effect of pulse frequency on microstructures and degradation behavior of electroformed zinc. Samples were produced by single-pulse electro-deposition technique with different pulse frequencies (50 Hz, 100 Hz, and 1000 Hz). By controlling the pulse frequency, the thickness of the zinc coating can be adjusted. The 50 Hz produced zinc film possesses strong (11.0) grain orientation, 100 Hz produced zinc film possesses clear ((11.0) and (10.0)) grain orientations, yet 1000 Hz produced zinc film shows more random grain orientations of (10.0), (10.1), and (11.0), which provides a possible way to design a controllable nanometer surface microtopography. Although thermodynamic degradation tendency implied from open current corrosion voltage were similar, the kinetic corrosion rate showed a clear increasing trend as pulse frequency increased from 50 Hz to 1000 Hz, which corresponded with the electrochemical impedance spectroscopy and long-term soaking test in hanks solution. According to ISO 10993-5:2009 and ISO 10993-4:2002, electrodeposited zinc materials produced in this study showed a benign hemolysis ratio and no toxicity for cell growth. Zinc prepared under 50 Hz shows the best blood compatibility. Electrodeposited zinc materials are expected to be used for the shell of a degradable dosing pump.

## 1. Introduction

Bioabsorbable metals and metal alloys are widely used in short-term implanted dosing pumps [1]. Zinc can be considered a candidate for the shell of implanted dosing pump because of its beneficial features [2]. Zinc is essential for normal basic functions of the body [3] and is also reported to be crucial to certain bio-functions such as immunity [4], osteogénesis [5], and keratinocyte migration [6]. In addition, zinc has specific and limited antibacterial properties [7]. Bowen has found that the degradation rate of pure zinc is suitable for clinical applications [8].

The continuous development of material processing technology and surface modification methods has promoted the research of medical devices with ideal mechanical, morphological and biocompatible [9] properties to meet functional requirements [10]. Appropriate coating of temporary biomedical implants could improve antibacterial properties [11], blood compatibility [12], osteogenesis, and tissue regeneration [13]. Materials commonly used to make coatings for biomedical implants include metals [14], polymers [15], and hydrogels [16]. Due to the low mechanical strength of pure zinc, it is not easy to fabricate a zinc layer with adjustable thickness with complex shape by conventional casting extrusion methods. Electroplating zinc can be used as a coating on the surface of objects to play a protective role [17]. However, most of these galvanized coatings are less than 5 μm in thickness. Electro-deposition is a rapid proto-typing thin-wall coating technology, which can prepare coating with a thickness of 10 μm to 5 mm according to the thickness requirements [18]. The principle of electro-deposition is similar to electroplating, in which the anode is oxidized to provide cations, which are deposited on the cathode. The purpose of electroplating technology is to make coatings for decoration and reinforcement. In contrast, electro-deposition technology aims to produce an independent entity and the grain size of electro-deposition products is much smaller than that of casting products [19].

Some studies have used electrodeposition technology to produce composite electrodeposition zinc coating to optimize its performance [20], but most of them use direct current electrodeposition [21]. Surface morphology is a significant factor affecting the biocompatibility of implanted medical devices. Youssef, KMS et al. [22] studied the surface morphologies of pulse current electrodeposited nanocrystalline zinc, but they did not investigate the effect of pulse frequency on surface morphology. For one thing, energy and nucleation rate determine the preferred orientation during crystallization, for another pulsed current of different frequencies can provide different amounts of energy. The pulse frequency is likely to affect the surface morphology of the implanted devices. At present, there is no research on the application of electrodeposition technology in the preparation of implanted drug pump shells. In this study, we hypothesized that changing the pulse frequency would result in different surface morphology, degradation rate, and blood compatibility. Therefore, we hope to explore the effect of pulse frequency on microstructures and the degradation behavior of electroformed zinc and find a possible way to design a controllable nanometer surface microtopography for the shell of an implantable dosing pump suitable for clinical applications.

In this study, we used pulsed current electro-deposition to prepare a bioresorbable zinc coating layer for the shell of an implantable dosing pump with a thickness from 50 μm to 500 μm. Samples were produced by the single-pulse electrodeposition technique with different pulse frequencies (50 Hz, 100 Hz, 1000 Hz). The coatings were characterized by X-ray diffraction analyses (XRD), scanning electron microscope (SEM), polarization test, electrochemical impedance spectroscopy (EIS), and long-term static immersion test. Then the cytocompatibility and hemolytic rate of electroformed zinc was evaluated. We focused on the influence of pulse frequency on the microstructure, mechanical, and corrosion properties of electroformed zinc material and analyzed and discussed the experimental results.

## 2. Materials and Methods

### 2.1. Materials Preparation

The composition of the electro-deposition solution and operation conditions are shown in Table 1. The schematic diagram of the experiment device is shown in Figure 1. On electro-deposition, a commercially available zinc specimen (99% purity, 60 mm × 150 mm × 3 mm) was used as the anode. A 10 mm × 10 mm titanium plate was used as the cathode for forming zinc coating. Due to the low mechanical strength of zinc, we designed the zinc film with a thickness of 400 μm. The thickness of the zinc coating was controlled by modulating the time for electro-deposition. In this study, all samples were electrolytically deposited for 7 h. After electro-deposition under unidirectional pulsed current at different frequencies (50 Hz, 100 Hz, 1000 Hz), all as-prepared samples were immersed in 1% HNO_3_ for about 2 s to remove the alkaline film on the surface of samples, then rinsed by deionized water and put into the natural-draft drier. Then the samples were carefully removed from the titanium electrode with a tweezer. The freshly prepared samples were used for SEM, XRD, polarization test, EIS, and long-term static immersion test.

### 2.2. Microstructure Characterization

The impact of different frequencies on grain orientation and morphology was analyzed using a scanning electron microscope (SEM, Quanta 250 FEG, FEI, USA) and X-ray diffraction analyses (XRD). XRD analyses were carried out using a Rigaku MiniFlex diffractometer, with Cu K_α_ radiation (1.54056 Å) to determine the grain size and the texture of the deposits. The scan rate was 0.02°/s over a 2*θ* ranging from 20° to 80°. The preferred orientation was calculated from the XRD spectrum. The preferred orientation of the obtained zinc samples was analyzed by Muresan’s method [23]. If any crystal plane shows a T_c_ result bigger than 1, zinc will have the preferred orientation on this crystal plane.

The average grain size could be calculated by the Scherrer equation [24] (Equation (1)):*D* = *Kγ*/*B*cos *θ*(1)
where *D* is the half-value breadth of the diffracted beam, *K* is a numerical constant that has the value of 0.93, *γ* is the X-Ray wavelength, *B* is the diffraction peak half-height width of the sample and *θ* is the Bragg angle.

### 2.3. Mechanical Test

The Vickers hardness of different pulse frequency electroformed zinc deposits (10 mm × 10 mm, mechanical polished) was conducted on Vickers micro hardness tester (HTV-PHS30, Foundrax, UK) at an applied load of 10 gf and indentation time of 15 s. As well, 6 measurements were taken at different positions on each zinc sample.

### 2.4. Static Immersion Test

The characterization of degradation behavior was measured by the immersion test. Four 5 mm × 5 mm zinc specimens were selected for each group prepared in different frequencies. Before the immersion test, all samples were ultrasonic cleaned in ethanol for 15 min and then in deionized water for 15 min. After drying the samples, weigh and record the weight of each sample. Then each zinc specimen was completely immersed in 50 mL of simulated body fluid Hank’s solution [25] at 37 °C. After 30 days, 60 days, and 120 days of immersion, samples were precisely measured the weight loss. The immersion degradation rate of zinc specimens was calculated according to Equation (2):*V* = (*q*_1_ − *q*_0_)/*Atρ*(2)
where *q*_1_ is the mass after immersion, *q*_0_ is the mass before immersion, *A* is the surface area of the specimen, *t* is the immersion time, and *ρ* is the density of zinc.

### 2.5. Electrochemical Measurements

The polarization curves were tested using a three-electrode system (Ag/AgCl as reference electrode, 10 mm × 10 mm zinc plate as working electrode, Platinum as an auxiliary electrode with a surface area approximately two times as zinc plate) with CHI 600B electrochemical workstation. The theoretical degradation rate of zinc specimens was calculated by Faraday’s Law [26]. The EIS test was then performed with the frequency ranging from 1 MHz to 10 KHz and analyzed its impedance data.

### 2.6. Cytocompatibility Evaluation

To investigate the feasibility of using electroformed zinc as the implantable materials, we accessed the cytocompatibility of electroformed zinc according to ISO10993-5:2009 [27]. Mesenchymal stem cells (MSC) and human umbilical vein endothelial cells (HUVEC) were chosen. For metal extract preparation, a metal plate with the geometric size of 10 mm × 10 mm was placed at the bottom of a 12-well flat-bottom plate, sterilized by UV light for 12 h. Then extracts were cultured for 24 h at 37 °C. Then we added 200 µL as-prepared metal supernatant into each well and cultured it with 1 × 10^4^ cells for 3 days. Finally, an MTT kit was used to detect cell activity. Three samples were taken from each group of different frequencies (50 Hz, 100 Hz, 1000 Hz) for cytocompatibility tests.

### 2.7. Hemolysis Evaluation

Diluted blood for hemolysis evaluation was prepared with healthy adult blood from a male volunteer. All the experiments were approved by the medical ethics committee of Xiangya III Hospital of Central South University (No. 2016-S139). Pure zinc samples were smashed and dipped in a 10 mL 0.9% NaCl solution for 30 min. A 0.9% NaCl solution was prepared as the negative control. Deionized water was prepared as the positive control. After incubation with 0.2 mL blood, zinc metals were removed, and the supernatant in each tube was measured by ultraviolet spectrophotometer at 545 nm. Three samples were taken from each group of different frequencies (50 Hz, 100 Hz, 1000 Hz) for hemolysis evaluation. The hemolysis result could be calculated using the formula as follows [28]:Hemolysis = (OD_test_ − OD_neg_)/(OD_pos_ − OD_neg_)(3)
where OD was the optical density at 545 nm.

### 2.8. Statistic Analysis

Data are presented as a mean ± standard deviation and a one-way analysis of variance (ANOVA) was employed with a Tukey’s HSD posthoc test using Graphpad version 7.4 software. A value of *p* < 0.05 was considered significant.

## 3. Results and Discussion

Figure 2 shows the theoretical and actual thicknesses in experiments of the electrodeposited sample. The *p* values between the calculated groups were all less than 0.05, indicating that the data differences between the groups were significant. With the increase of pulse frequency from 50 Hz to 1000 Hz, the thickness of the electrodeposited zinc layer decreased from 396 μm to 363 μm. Higher pulse frequency leads to a lower energy conversion rate, and therefore lower zinc formation efficiency. Such phenomenon is due to the zinc deposition kinetic processes during pulse electro-deposition. The primary technique adopted in this experiment is single-pulse electro-deposition. During one pulse cycle, charging time and pulse duration time inverse the pulse frequency. The double electric layer is formed during charging and acts as a resistance at the beginning of each pulse. According to the capacitance effect theory mentioned by Gülesin Yılmaz [29], the charging time and discharging time are much shorter than the whole pulse duration, and owing to the increasing pulse frequency, the considerable accumulation of capacitive effect reduces the energy efficiency. Therefore, the energy efficiency decreases correspondingly as pulse frequency increase, thus making the thickness of electro-deposition zinc decrease with the increase of pulse frequency.

The XRD patterns identified the samples electrolytically deposited under 50 Hz, 100 Hz, and 1000 Hz pulse frequency as pure zinc (PDF NO.04-0831) with a hexagonal close-packed (hcp) crystal structure in Figure 3. The average grain size of the coating is estimated by the Scherrer equation, which is 641 Å in 50 Hz, 457 Å in 100 Hz, and 387 Å in 1000 Hz. Moreover, all the samples exhibited considerably high crystallinity with the remarkably high sharpness of the XRD patterns.

To explore the preferred orientation of the electrodeposited zinc deposit, the texture coefficient was calculated according to Muresan’s method [30]. The corresponding texture coefficient (Tc) values are shown as a bar chart in Figure 4. According to Muresan’s method, the Tc value above 1 refers to the preferred orientation of a certain plane. Figure 4 shows that (11.0) and (10.0) are the preferred orientation for all frequencies.

The atom density of the (11.0) plane is relatively higher than that of the (10.0) or (10.1) plane in hcp structure, and the surface energy of the (10.0) or (10.1) plane is lower than that of the (11.0) plane [31]. The Tc value of (11.0) at 50 Hz is much higher than 100 Hz and 1000 Hz. When frequency arises from 50 Hz to 1000 Hz, the Tc value of (10.1) and (10.2) increases, while the Tc value of (11.0) and (10.3) decreases. During crystallization, energy and nucleation rate determine the preferred orientation. In the normal crystallization process, where energy dominate, atoms always tend to crystallize in the (00.2) plane due to the lowest surface energy. However, the electroplating process provides sufficient energy, and the nucleation rate becomes more important. If the increased fresh metal atoms do not migrate to the (10.0) plane, then some of these atoms could stay at the (11.0) plane thus the orientation is changed. As mentioned above, a lower frequency means a relatively longer T_off_ and T_on_, and according to the capacitive effect, more energy is charged to the samples under 50 Hz. Then a more rapid nucleation rate exists under 50 Hz. Finally, the (11.0) plane becomes dominant for the samples electrolytically deposited under 50 Hz. But such an effect is less obvious for the 1000 Hz sample as less energy is inputted to the system, which makes zinc film exhibit more random orientations at 1000 Hz [32] and the Tc value of the (11.0) plane decreased. This finding could be further studied to control desirable surface conditions for the implantable electronic device material.

Figure 5 shows the SEM images of the surface morphology of zinc films which are electrolytically deposited under 50 Hz, 100 Hz, and 1000 Hz pulse frequency. The electrodeposited zinc materials all showed metallic brightness and smooth surface observed with naked eyes, but when magnified by SEM, the samples exhibited an anisotropic dendritic morphology. Figure 5 represents grain obtained under 1000 Hz pulse frequency tends to grow more randomly compared to zinc film under 50 Hz and 100 Hz, which is consistent with the Tc results in Figure 4. Both XRD and SEM results indicate that frequency has effects on the grain orientations for electrodeposited zinc, and in our case, reducing PC frequency would help to grow the preferred orientation in zinc, and versa.

Figure 6 shows the SEM images of the cross-section of deposited coatings under 100 Hz and 1000 Hz. The grain orientation can not be seen from the cross-section image, but it can be seen that the matrix formed by electrodeposition is very dense, which can effectively prevent the infiltration of water and water vapor.

Higher hardness of coating for implantable electronic devices is desirable, as it secures the electrodeposited layer structural functionality. Figure 7 shows the bar chart of Vickers hardness of pure electrodeposited zinc under different PC frequencies. The microhardness of the three samples is almost the same. In addition, electrodeposited zinc shows better performance (~48 HV) than normal zinc (37 HV) [33].

The corrosion properties could be detected by Tafel curves, where the E_corr_ and I_corr_ are the most important parameters which can serve as theoretical predictions. Figure 8 shows the Tafel plot test of three samples carried out in CHI660B electrochemical station. It can be seen from Table 2, the calculated Tafel slope reveals that zinc under 50 Hz shows a slightly lower Tafel slope than zinc under 100 Hz and 1000 Hz, which indicates the lower corrosion rate of 50 Hz zinc due to the higher activation energy, though the difference is not significant. Table 2 lists the annual corrosion rate calculated by Faraday’s law. E_corr_ of zinc under 50 Hz, 100 Hz, and 1000 Hz are close, which indicates the similar corrosion tendency of zinc prepared under different PC frequencies. As the pulse frequency changes from 50 Hz to 1000 Hz, the annual corrosion rate gradually increases from 0.11182 mm/year·cm^2^ to 0.12779 mm/year·cm^2^, which shows a non-significant increasing trend and is similar to as-cast pure Zinc (0.132 mm/year·cm^2^) [26].

After the electrochemical test, a long-term simulation degradation test should be conducted to testify to the degradation property of certain materials. We conducted immersion tests for up to 120 d in Hank’s solution. As indicated in Figure 9, the degradation rate of zinc samples electrolytically deposited under a pulse frequency of 1000 Hz is higher than zinc samples electrolytically deposited under 100 Hz and 50 Hz. A positive correlation between degradation rates and immersion time is found. All electrodeposited zinc samples show accelerated degradation behavior. Interestingly, this phenomenon is more apparent when the immersion time changes from 30 days to 60 days, which means the degradation rate accelerated with prolonged soaking time, despite the corrosion product layer formed during corrosion. However, only the 30-day degradation rates are lower than bulk zinc (0.0438 mm year^−1^) [34], and the 60-day and 120-day degradation rates are similar to the literature values.

To further address the difference in degradation rate, as shown in Figure 8 and Figure 9, EIS measurements in Hank’s solution (37 °C) were conducted. The impedance diagram is shown as a Nyquist plot in Figure 10. The equivalent circuit fitted to study impedance spectra is shown in Figure 11. Mou Cheng Li et al. [35] obtained a similar equivalent circuit of zinc in 3.5% NaCl solutions. Table 3 indicates the impedance parameters.

The zinc sample electrolytically deposited under 50 Hz pulse possesses higher Rp and lower double-layer capacitance than zinc samples under 100 Hz and 1000 Hz, the Rp value increases in the order 1000 Hz < 100 Hz < 50 Hz. Thus, the order of samples corrosion resistance is 50 Hz < 100 Hz < 1000 Hz, which is coincidental with Tafel plots. The impedance data from Table 3 indicates the significant improvement of polarization resistance on 1000 Hz zinc samples after the static immersion test. EDAX results show the increased concentration of O and the existence of elements C and Zn on the 1000 Hz static immersed zinc sample. From SEM characterization and L Yin’s work [34], we understood that upon being immersed in Hank’s solution, zinc quickly reacts with the acid components, forming a porous layer consisting of ZnO, Zn(OH)_2_, and other chloride corrosion products [35].

The surface morphologies of both original and soaked zinc films (1000 Hz) in Figure 12 clearly illustrate the effect of corrosion on surface conditions. Figure 12a shows some pseudo-hexagonal plane crystals on the zinc coating samples. The white porous corrosion product layer on the metal substrate during immersion is observed in Figure 12b. After the immersion test, the crystals grow and form islands that spread over the entire surface.

The difference in degradation rate could be explained by the electro-deposition process of zinc. Girin [31] and Jantaping [36] reported that zinc coatings with a prismatic (11.0) texture have better corrosion resistance than other coatings. According to their study, the zinc layer with (11.0) texture promotes the formation of the amorphous-oxide layer, and this formed amorphous-oxide layer plays an effective protective barrier role which could effectively reduce the corrosion of zinc samples. In the corrosive environment, the corrosion rate of each zinc metal grain varies because of the difference in the binding energy of atoms between the crystallographic planes [37]. Mouanga successfully inhibited the corrosion of zinc by using urea; the presence of urea leads to the increase of (11.2) plane’s intensity [38]. In our study, zinc film prepared under 50 Hz pulse frequency showed obviously (11.0) preferred orientation, which may be the reason for the better anti-corrosion performance of 50 Hz-zinc than the others.

As shown in Figure 13, in the cytocompatibility evaluation, the toxic effects on the cell viability of MSC are higher than those on HUVEC. The toxic effects on the cell viability of all the electrodeposited zinc samples do not significantly differ. S-295 brightener was used in the preparation process of electrolytic deposition, and there was inevitable residue. However, the cytocompatibility evaluation results indicate that electrodeposited zinc materials do not present any toxic effects on MSC and HUVEC cells and exhibit excellent biocompatibility, which is in good consistent with the results of coating fabricated by the conventional casting methods [39].

Table 4 shows the results of the hemolysis test. The calculated hemolysis rates of zinc (50 Hz, 100 Hz, and 1000 Hz) are 0.478%, 1.368%, and 0.752%, respectively, which are far less than the safe value of 5%, suggesting that the electrodeposited zinc materials would not lead to severe hemolysis according to ISO 10993-4:2002. Among them, zinc prepared under 50 Hz shows the best blood compatibility. One possible explanation is that the microphase structure with certain roughness can sometimes obtain extremely high blood compatibility, and the material with heterogeneous microphase structure in the range of 0.1~2 μm has better anticoagulant properties [40]. According to Figure 5, the surface grain size of zinc under 50 Hz is 0.628 ± 0.095 μm, and the surface grains show an anisotropic arrangement and heterogeneous microfacies structure.

## 4. Conclusions

In this study, the way to control the thickness of zinc films by electro-deposition time management has been found and used to produce 50 μm~500 μm zinc films, successfully. Pulse electrodeposited zinc materials with different preferred grain orientations and microstructures were produced by varying pulse frequencies to 50 Hz, 100 Hz, and 1000 Hz. The main conclusions are as follows.

The thickness can be adjusted by electrodeposition, which is convenient and controllable, and the product has no cracking phenomenon.The test results also imply that pulse frequency will affect the grain orientation, and thus the corrosion properties. It is shown that the 50 Hz produced zinc film possesses strong (11.0) grain orientation, 100 Hz produced zinc film possesses clear (11.0) and (10.0) grain orientations, yet 1000 Hz produced zinc film shows more random grain orientations of (10.0), (10.1) and (11.0).The effect of the pulse frequency resulting from microstructures was clarified by electrochemical tests. Although thermodynamic degradation tendency implied from open current corrosion voltage (E_corr_) were similar, the kinetic corrosion rate showed a clear increasing trend as pulse frequency increased from 50 Hz to 1000 Hz, which corresponded with the EIS test and long-term soaking test in hanks solution. This tendency is probably attributed to the refined grain that increased the structural stability of the PC-formed zinc. It provides a possible way to design a controllable nanometer surface microtopography by adjusting PC frequency.Our results also indicated that appropriate pulse frequency can improve the blood compatibility of the material. According to ISO 10,993-5:2009 and ISO 10993-4:2002, electrodeposited zinc materials produced in this study showed a benign hemolysis ratio and no toxicity for cell growth. Zinc prepared under 50 Hz shows the best blood compatibility. Electrodeposited zinc materials are expected to be used for the shell of a degradable dosing pump.

## Figures and Tables

**Figure 1 bioengineering-09-00289-f001:**
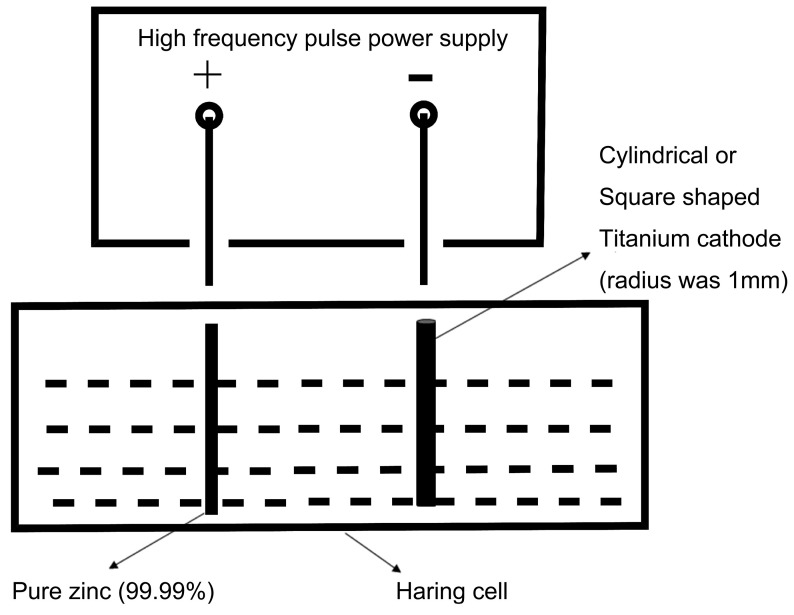
The schematic diagram of the experiment device.

**Figure 2 bioengineering-09-00289-f002:**
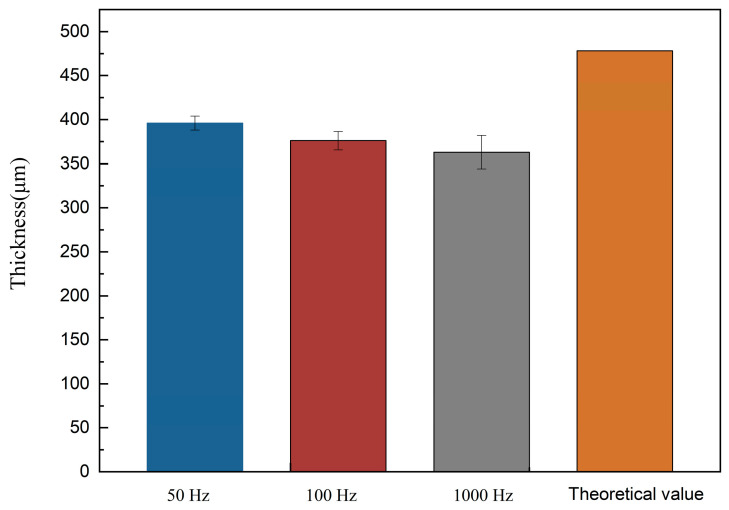
The theoretical and real thickness of electroformed zinc.

**Figure 3 bioengineering-09-00289-f003:**
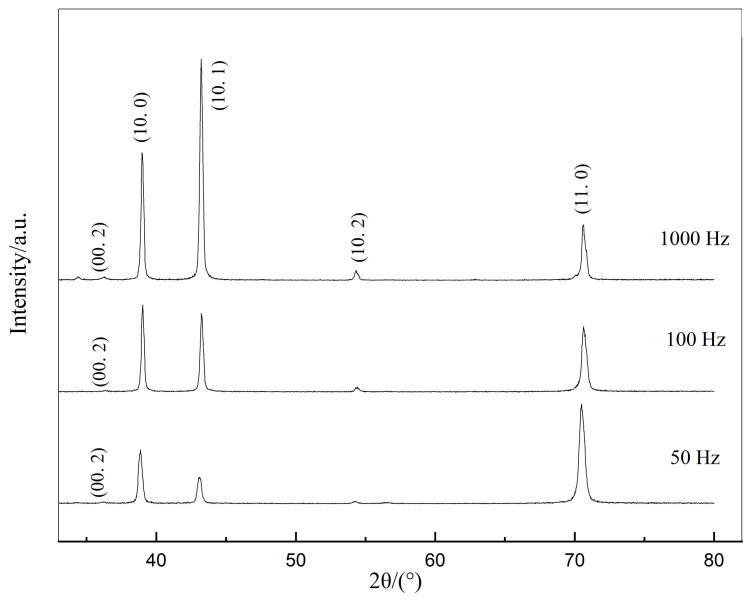
XRD patterns of electrodeposited zinc under 50 Hz, 100 Hz, and 1000 Hz pulse frequency.

**Figure 4 bioengineering-09-00289-f004:**
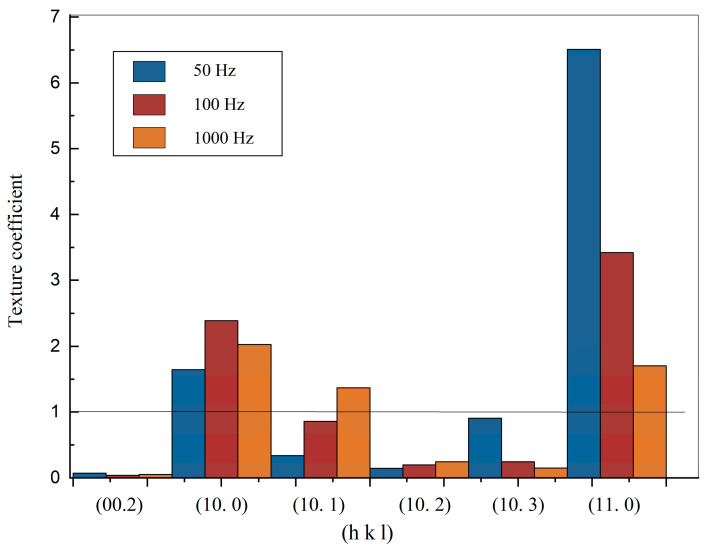
Tc chart of electrodeposited zinc under 50 Hz, 100 Hz, 1000 Hz pulse frequency.

**Figure 5 bioengineering-09-00289-f005:**
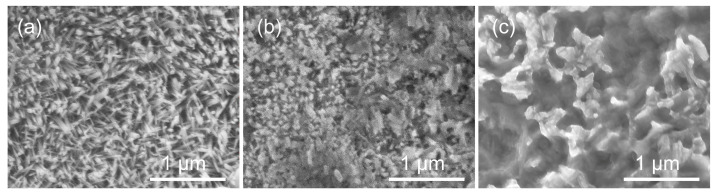
SEM of electrodeposited zinc film: (**a**–**c**) are under 50 Hz, 100 Hz, and 1000 Hz, respectively.

**Figure 6 bioengineering-09-00289-f006:**
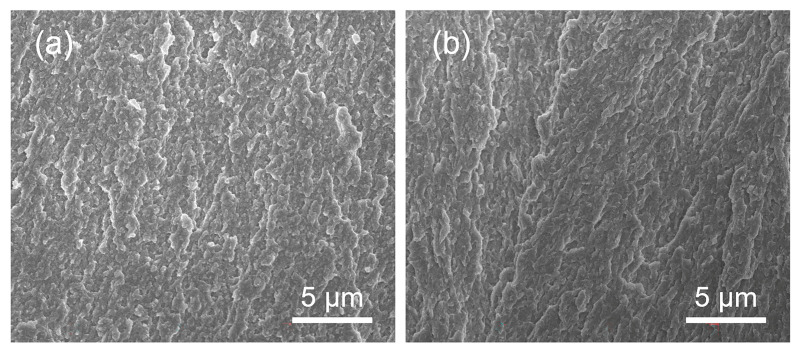
SEM of the cross-section of deposited coating: (**a**,**b**) are under 100 Hz and 1000 Hz, respectively.

**Figure 7 bioengineering-09-00289-f007:**
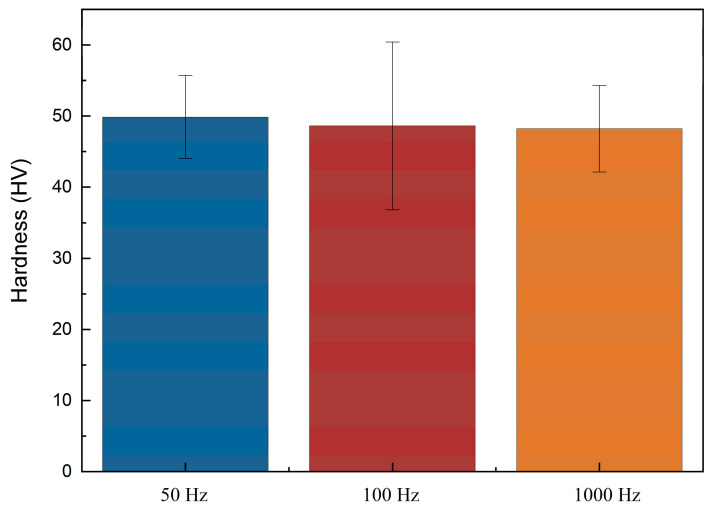
Bar chart of Vickers hardness of electrodeposited zinc under 50 Hz, 100 Hz, and 1000 Hz.

**Figure 8 bioengineering-09-00289-f008:**
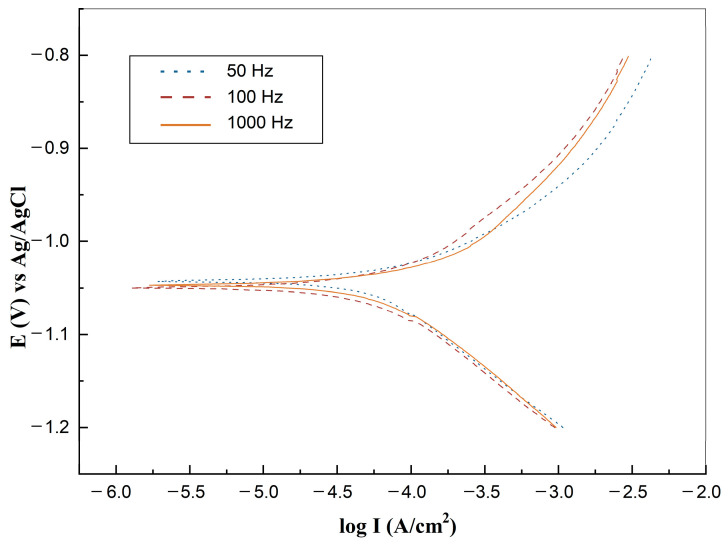
Tafel plot of zinc under 50 Hz, 100 Hz, and 1000 Hz, respectively (37 °C immersed in Hank’s solution for 30 min).

**Figure 9 bioengineering-09-00289-f009:**
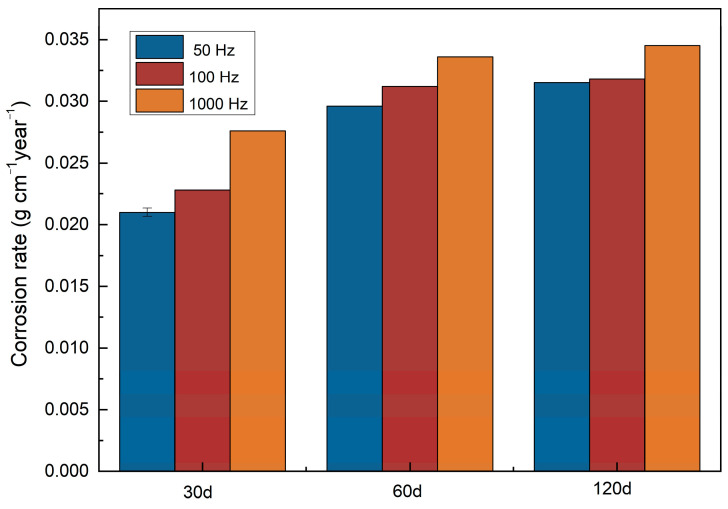
Degradation rates of electrodeposited zinc for 30, 60, and 120 days.

**Figure 10 bioengineering-09-00289-f010:**
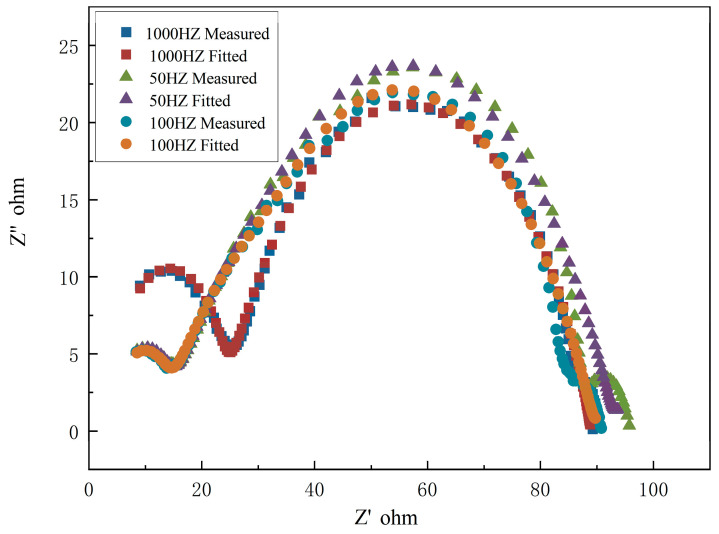
Nyquist plot of zinc under 50 Hz, 100 Hz, and 1000 Hz.

**Figure 11 bioengineering-09-00289-f011:**
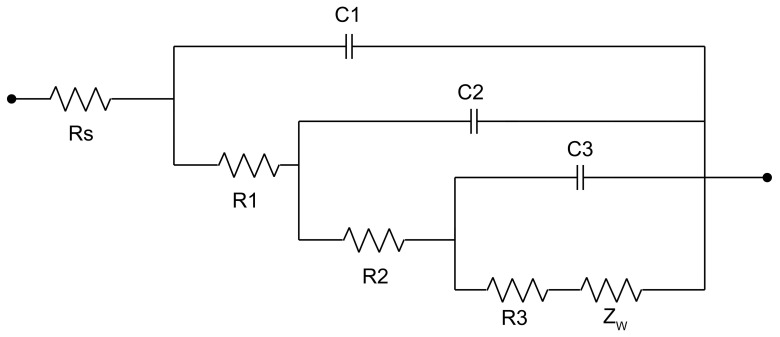
The equivalent circuit used to fit the Nyquist plot in Figure 9.

**Figure 12 bioengineering-09-00289-f012:**
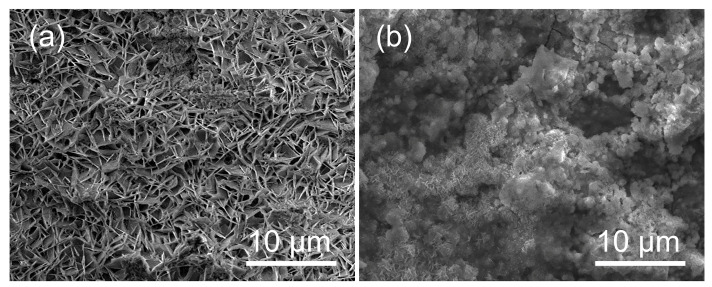
SEM of zinc film after electrochemical impedance studies: (**a**) 1000 Hz zinc before static immersion test. (**b**) 1000 Hz zinc after 30 day’s static immersion test.

**Figure 13 bioengineering-09-00289-f013:**
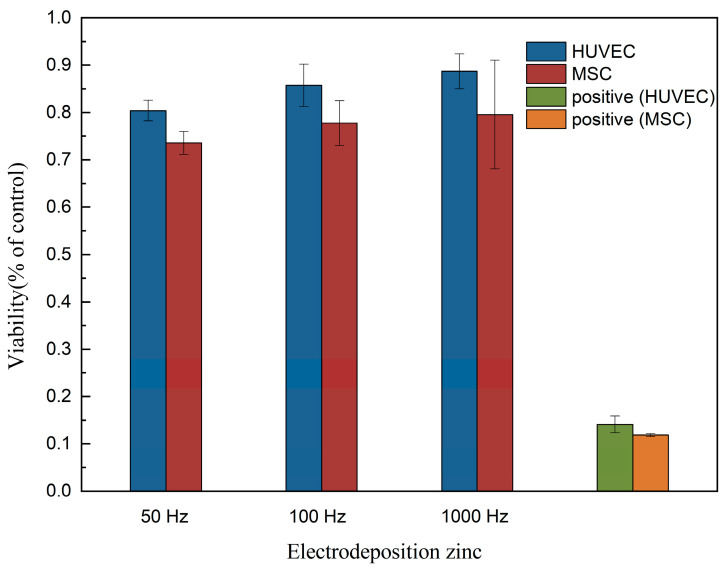
The cytotoxicity of electrodeposited zinc.

**Table 1 bioengineering-09-00289-t001:** The composition of the electro-deposition solution and operation conditions.

Bath Composition	Range	Operation Conditions
ZnSO_4_·7H_2_O	300–450 g/L	300 g/L γ = 30%
H_3_BO_3_	25–35 g/L	25 g/L t = 7 h
Brightening agent (S-Z95)	18–20 mg/L	19 mg/L
pH	3.5–5.5	4
Temperature	283–323 K	313 K
Cathode-current density	1–4 A/dm^2^	4 A/dm^2^
		Ton/Toff = 2/1

**Table 2 bioengineering-09-00289-t002:** Electrochemical parameters of zinc under 50 Hz, 100 Hz, and 1000 Hz, respectively.

Samples (Hz)	βa (mV/decade)	βb (mV/decade)	Icorr (10^−6^ A/cm^2^)	Ecorr (V)	C (mm∙year^−1^∙cm^2^)
50	15.148	8.118	7.993	−1.0427	0.11182
100	16.155	8.334	8.134	−1.0497	0.12092
1000	16.388	8.445	8.596	−1.0474	0.12779

**Table 3 bioengineering-09-00289-t003:** Impedance parameters.

F (Hz)	Rs (Ω·cm^2^)	C1 (μF·cm^−2^)	R1 (Ω·cm^2^)	C2 (μF·cm^−2^)	R2 (Ω·cm^2^)	C3 (μF·cm^−2^)	R3 (Ω·cm^2^)	Rp * (Ω·cm^2^)
50	4.664	0.1916	10.22	184.1	49.65	41.89	28.45	88.32
100	4.425	0.1849	9.857	21.84	48.21	57.28	27.12	85.19
1000	4.582	0.1301	8.82	135.7	47.57	82.59	27.8	83.65

* Polarization resistance (RP) is the sum of all three resistances R1, R2, and R3.

**Table 4 bioengineering-09-00289-t004:** The cytotoxicity of electrodeposited zinc.

	50 Hz	100 Hz	1000 Hz	Positive Control	Blank Control
OD (A)	0.043	0.056	0.047	0.036	1.498
T (%)	90.7	87.7	89.7	91.8	3.1
Hemolysis rate	0.478	1.368	0.752	-	-

OD: optical density; T: transmittance.

## Data Availability

The data that support the findings of this study are available from the corresponding author upon reasonable request. All figures in this paper are original.

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
