# Peer review of "Effect of Pulse Frequency on the Microstructure and the Degradation of Pulse Electroformed Zinc for Fabricating the Shell of Biodegradable Dosing Pump"

_bioengineering, 2022, doi:10.3390/bioengineering9070289_

Round 1

Reviewer 1 Report

Dear Authors,

Thank you for interesting paper. After reading the presented manuscript, I have some questions and suggestion as below.

1. Why the as-prepared samples were immersed in 1% HNO3 solution? What was the aim of this process?

2. Was the electrolyte bath stirred during deposition process?

3. Line 74 - the word "electronic" and "Quantan" should be corrrected to the form "electron" and "Quanta", respectively.

4. The Scherer equation to calculate of average grain size should be provided.

5. Line 86/87 - What does the "30d, 60d, 120d" mean? Probably, you mean about time (in second) of immersion. Please, correct it.

6. The designation markings of equation number 1 and 2 are not explained. Please, add it.

7. How did you measured the Vickers hardness? The thickness of deposited coating was relatively thin with respect to the applied indenter. Are you sure that obtained results relate to the zinc coating and not the substrate?

8. Did you analysed the bright particles on the surface of deposited coatings (especially visible in figure 5c)? What are that?

The microstructure observations should be extended to the cross-section of deposited coating.

Best regards,

Reviewer

Author Response

Many thanks for your helpful feedback. We revised our manuscript carefully according to Reviewers’ Comments.All the questions the reviewers brought forward were answered in detail. Please see the attachment.

Reviewer 2 Report

Comments to the author:

This manuscript reports on showing the significant effect of pulse frequency during applying the electrodeposition method on grain orientation of the final coating.

There are quite a few works needed overall on building the motivation in the introduction and application in the conclusion. There are concerns with the presentation of data in the figures, and the explanation of data is not thorough.

1. The abstract should contain the best data from the paper. Specific data has to be included in the abstract. The significance of the study also needs to be mentioned in the abstract.

2. Writing of the overall context needs to be improved and smoothed for flow.

3. More details about the SEM and XRD parameters need to be supplemented in section 2.2.

4. When reporting on animal or human experiments, a statement that ethical approval from the national or local authorities was obtained should be added to the main text of your manuscript, including the assigned approval/accreditation number.

5. Page 4, line 119: please add statistical analysis for comparing the coating thickness under different conditions.

6. Page 5, line 140. Reference needs to be added here: To explore the preferred orientation of electrodeposited zinc deposit, the texture co-139 efficient was calculated according to Muresan’s method.

7. Page 5, lines 146-149: Why did there happen to be a drop in the Tc value of the (11.0) plane compared to the (10.0) plane in the 1000 Hz group?

8. Page 6, line 175: please check the figure caption of Figure 5.

Author Response

(The authors gave the same response as above.)

Reviewer 3 Report

Page 1, Section 1. Please note the past work on pulsed deposition that showed finer grain sizes and improved corrosion resistance: Kh.M.S Youssef, C.C Koch, P.S Fedkiw, Improved corrosion behavior of nanocrystalline zinc produced by pulse-current electrodeposition, Corrosion Science, Volume 46, Issue 1, 2004, Pages 51-64. https://doi.org/10.1016/S0010-938X(03)00142-2

Page 3 line 78: “Scherrer”, please correct the spelling.  Also, it is very easy to write the Scherrer equation, I recommend including it. At minimum, please give a reference to find this equation, for example Patterson, A. (1939). "The Scherrer Formula for X-Ray Particle Size Determination". Phys. Rev. 56 (10): 978–982. Bibcode:1939PhRv...56..978P. doi:10.1103/PhysRev.56.978.

Page 3 line 86: Please give a reference for Hank’s solution such as the original Hanks JH, Wallace RE (1 June 1949). "Relation of oxygen and temperature in the preservation of tissues by refrigeration" (PDF). Proceedings of the Society for Experimental Biology and Medicine. New York, N.Y.: Society for Experimental Biology and Medicine. 71 (2): 196–200. PMID 18134009.

Please define the terms in Eq. 1

Page 5, line 140, please give a reference for Muesan’s method, such as the original L. Muresan, L. Oniciu, M. Froment, G. Maurin. Electrochim. Acta, vol. 37 (1992), p. 2249.

Figure 12 Meaningful figure caption is missing

Define terms used in Figure 12: HUVEC, MSC…These terms ae never defined in the text, and are used in both figure and text.

Page 10 line 280 “Conclusions” please correct the spelling

Author Response

(The authors gave the same response as above.)

Reviewer 4 Report

The reviewer comments of the paper «Effect of Pulse Frequency on the Microstructure and the Degradation of Pulse Electroformed Zinc for Fabricating the Shell of Biodegradable Dosing Pump» - Reviewer

The authors presented an article «Effect of Pulse Frequency on the Microstructure and the Degradation of Pulse Electroformed Zinc for Fabricating the Shell of Biodegradable Dosing Pump». However, there are several points in the article that require further explanation.

Comment 1:

The abstract needs to be improved.

Demonstrate in the abstract novelty, practical significance. All abbreviations appearing in the article for the first time should be explained. In particular, what is a PC?

Comment 2:

The introduction needs to be improved.

Firstly, group quotation is unacceptable in one phrase, for example [5-8], [11-13]. Break this sentence into parts or individual sentences. For example, ... [...], ... [...], etc. Or one reference - one sentence.

Now the list of references needs to be supplemented with at least 6-8 more references published over the past 5 years. Here are some recent articles:

Coatings 2021, 11, 712. doi:10.3390/coatings11060712

International Journal of Bioprinting 2022, 8(1), 74-95. doi:10.18063/IJB.V8I1.501

Materials 2021, 14, 2253. doi:10.3390/ma14092253

After analyzing the literature, show before formulating the goal of the "blank" spots. Which has not been previously done by other researchers. You must show the importance of the research being undertaken. Show what will be the new research approach in this article. You need to show a hypothesis.

Add scientific novelty and practical significance to the article.

Add a clear purpose to the article.

Briefly describe what is done in each section of the article.

Comment 3:

2. Materials and Methods

Are all figures original? If not needed appropriate citations and permissions. Refine this for figures throughout the article.

The quality and resolution of graphic abstract needs to be improved. Now they are vague and not clear to the reader.

Describe the measurement procedure in more detail. At what point in time? How is the measuring setup set up? How many repetitions of measurements? What statistical methods are used to process experimental results? Describe the experimental stand in more detail. What method of experiment planning is used and why?

Comment 4:

4. Results and discussion

Add in caption cutting conditions for which each figure is obtained in the captions.

The description of all figures in the text must be supplemented. Minimum 4-5 sentences. Compare these results with previously published references. Provide relevant citations.

Redraw the figures 2,4,6,7,8,9,12 in color.

Comment 5:

Conclusions needs to be improved.

It is necessary to more clearly show the novelty of the article and the advantages of the proposed method. Add qualitative and quantitative results of your work. What is the difference from previous work in this area? Show practical relevance.

Use the format for a few paragraphs:

• Conclusions 1

• Conclusions 2

• Etc.

The article is interesting, but needs to be improved. Authors should carefully study the comments and make improvements to the article step by step. After major changes can an article be considered for publication in the "Bioengineering".

Author Response

Many thanks for your helpful feedback. We revised our manuscript carefully according to Reviewers’ Comments. All the questions the reviewers brought forward were answered in detail. Please see the attachment.

Round 2

Reviewer 2 Report

Dear Authors, thanks for your revisions. You appear to have addressed all of my comments in point-by-point answers in your replies to the reviewers. The authors well addressed the most issues and the manuscript is now suitable for publication.

Author Response

Dear reviewer, thanks again for your helpful feedback.

Reviewer 4 Report

The authors have improved the article. However, some comments should be explained more clearly. Another revision is needed.

1. The abstract needs to be improved.

Demonstrate in the abstract novelty, practical significance. All abbreviations appearing in the article for the first time should be explained. What is XRD, etc.?

2. The introduction needs to be improved.

Firstly, group quotation is unacceptable in one phrase, for example functions [5, 6, 7, 8], [11, 12, 13, 14, 15], [16, 17, 18]. One reference - one sentence.

After analyzing the literature, show before formulating the goal of the "blank" spots. Which has not been previously done by other researchers. You must show the importance of the research being undertaken. Show what will be the new research approach in this article. You need to show a hypothesis. 

Add scientific novelty and practical significance to the article.

Add a clear purpose to the article.

Briefly describe what is done in each section of the article.

3. 2. Materials and Methods

Are all figures original? If not needed appropriate citations and permissions. Refine this for figures throughout the article.

The quality and resolution of graphic abstract needs to be improved. Now they are vague and not clear to the reader.

Describe the measurement procedure in more detail. At what point in time? How is the measuring setup set up? How many repetitions of measurements? What statistical methods are used to process experimental results? Describe the experimental stand in more detail. What method of experiment planning is used and why?

Author Response

Many thanks for your helpful feedback. We revised our manuscript carefully according to Reviewers' Comments. All the questions the reviewers brought forward were answered in detail. Please see the attachment.
